# Experiential Avoidance in Primary Care Providers: Psychometric Properties of the Brazilian “Acceptance and Action Questionnaire” (AAQ-II) and Its Criterion Validity on Mood Disorder-Related Psychological Distress

**DOI:** 10.3390/ijerph20010225

**Published:** 2022-12-23

**Authors:** Tatiana Berta-Otero, Alberto Barceló-Soler, Jesus Montero-Marin, Shannon Maloney, Adrián Pérez-Aranda, Alba López-Montoyo, Vera Salvo, Marcio Sussumu, Javier García-Campayo, Marcelo Demarzo

**Affiliations:** 1Centro Mente Aberta (Brazilian Center for Mindfulness and Health Promotion), Department of Preventive Medicine, UNIFESP, São Paulo 04753-060, Brazil; 2Institute of Health Research of Aragon (IIS), 50009 Zaragoza, Spain; 3Primary Care Prevention and Health Promotion Research Network (RedIAPP), 28029 Madrid, Spain; 4Department of Psychiatry, Warneford Hospital, University of Oxford, Oxford OX3 7JX, UK; 5Teaching, Research & Innovation Unit, Parc Sanitari Sant Joan de Déu, 08830 Sant Boi de Llobregat, Spain; 6Consortium for Biomedical Research in Epidemiology & Public Health (CIBER Epidemiology and Public Health—CIBERESP), 28029 Madrid, Spain; 7Department of Basic, Clinical Psychology, and Psychobiology, Universitat Jaume I, 12006 Castellón de la Plana, Spain; 8Psychiatric Service, Hospital Miguel Servet, 50009 Zaragoza, Spain; 9Department of Psychiatry, University of Zaragoza, 50009 Zaragoza, Spain

**Keywords:** AAQ-II, primary care providers, psychological flexibility, experiential avoidance, self-criticism, mindfulness, anxiety, depression

## Abstract

Background: A sizeable proportion of Brazilian Primary Care (PC) providers suffer from common mental disorders, such as anxiety and depression. In an effort to cope with job-related distress, PC workers are likely to implement maladaptive strategies such as experiential avoidance (EA). The Acceptance and Action Questionnaire (AAQ-II) is a widely used instrument that evaluates EA but has shown questionable internal consistency in specific populations. This study assesses the psychometric properties of the AAQ-II among Brazilian PC providers, evaluates its convergence and divergence with self-criticism and mindfulness skills, and explores its criterion validity on anxiety and depressive symptoms. Methods: A cross-sectional design was conducted in Brazilian PC services, and the sample included 407 PC workers. The measures evaluated EA, self-criticism, mindfulness, depression, and anxiety. Results: The one-factor model of the AAQ-II replicated the original version structure. The AAQ-II presented good internal consistency among Brazilian PC providers. A multiple regression model demonstrated higher relationships with self-criticism than mindfulness skills. The criterion validity of the AAQ-II on anxiety and depression was stronger in the context of more severe symptoms. Conclusions: The AAQ-II is an appropriate questionnaire to measure the lack of psychological flexibility among Brazilian PC workers in the sense of EA.

## 1. Introduction

Psychological flexibility is associated with the ability to pursue valued goals despite being in the presence of troublesome experiences [1]. This construct has been proposed as a process of change underlying “Acceptance and Commitment Therapy” (ACT) [2], which is a contextual therapy formed by principles such as acceptance, cognitive defusion, values-based activation, and persistence. The principles of ACT work in unison to help increase the capacity to adapt to the circumstances of life without losing sight of one’s values and goals. One of the most widely used self-report instruments to assess psychological flexibility is the “Acceptance and Action Questionnaire” (AAQ-II) [3], which considers acceptance and action as key experiential components. The AAQ-II includes seven items that measure experiential avoidance (EA), which is used as an example of psychological inflexibility. EA is operationalized as the unwillingness to experience painful thoughts and emotions and is associated with the inability to be present and behave according to one’s value-directed actions [4]. It has been said that EA may be implicated in the etiology of various forms of psychopathology and that recovery of mental illness might be mediated through reductions in EA [5]. As mentioned above, this concept was originally developed within the ACT framework. Still, it seems to apply to other forms of “third wave” Cognitive-Behavioral Therapies (CBTs), such as “Dialectical Behavior Therapy” (DBT) [6] or “Attachment-Based Compassion Therapy” (ABCT) [7], among others.

The AAQ-II is the second iteration of the questionnaire. It was developed to overcome some psychometric problems observed in the previous version related to poor internal consistency values and an unstable factorial structure [8]. Despite the AAQ-II seeming to have overcome the lack of refinement in operationalizing EA, even after being translated into different languages, it still demonstrates questionable internal consistency values in specific populations. Content-specific versions have been created to increase psychometric properties in target populations and contexts. For instance, some modified versions of the AAQ-II have been adapted for the workplace (Work-related Acceptance and Action Questionnaire, WAAQ) [9]; for adults with hearing loss (Acceptance and Action Questionnaire-Adult Hearing Loss (AAQ-AHL)) [10]; for individuals who are overweight or obese (Food Craving Acceptance and Action Questionnaire, FAAQ)) [11,12]; for adults with chronic pain (Acceptance and Action Questionnaire II-pain version, AAQ-II-P) [13]; and cardiac patients (Cardiovascular Disease Acceptance and Action Questionnaire (CVD-AAQ)) [14]. A previous exploratory study that adapted a Portuguese version of the AAQ-II, using a sample of Brazilian college students, presented a unidimensional structure of the AAQ-II with a general factor having promising psychometric properties [15]. However, this study had some methodological shortcomings. For example, this study only used exploratory factorial analysis (EFA) and did not make use of confirmatory factorial analysis (CFA). Moreover, this paper did not evaluate the fit of the unidimensional model proposed and did not provide internal consistency estimates according to the most recent standards. The external validity of this study was also limited due to the sample being restricted to college students. Therefore, it is necessary to carry out subsequent studies, looking at specific target population samples, using this adapted version of the AAQ-II to help evaluate its factorial validity in the Brazilian population.

Primary Care (PC) providers constitute a specific target population since they are highly affected by job-related mood-disorder psychological distress [16]. In Brazil, it has been highlighted that 21% of PC workers might suffer from common mental disorders, such as anxiety and depression [17]. PC providers could be using EA as an escape mechanism to cope with their sense of lack of control over stressors in their current working environments [18]. In fact, it has been proposed that, for example, EA is a significant predictor of burnout among psychology and nursing undergraduate students [19].

Two other variables have been identified as significantly associated with distress and burnout in similar populations: self-criticism and mindfulness [20,21,22]. Self-criticism, as the negative counterpart of self-compassion [21], can be considered a dysfunctional emotion regulation strategy [23,24] used to deal with negative events in which one views their own mistakes as the cause of an event [25]. On the other hand, mindfulness refers to a mental trait or state associated with increased awareness of the present moment and characterized by a non-judgmental attitude. Past research has indicated that mindfulness skills are associated with several indicators of physical and psychological health [26,27]. Reducing self-criticism and improving mindfulness skills may help prevent and reduce psychological distress through the process of breaking patterns of reactivity that trigger anxiety and depression, and the cycle of suffering [28]. Both the presence of self-criticism and the absence of mindfulness skills can be addressed through the practice of meditative exercises that promote attention to the present moment [29]. It has been observed that mindfulness practices, for example, in the context of mindfulness-based programs (MBPs), may improve anxiety, depression, psychological distress, and mental well-being in healthcare professionals [30]. Moreover, past work suggests that these MBPs may improve outcomes through the mediating role of EA, self-criticism, and mindfulness [31]. Previous studies have shown significant relationships between burnout, EA, and self-criticism, as well as significant and negative relationships between burnout and the presence of mindfulness skills in healthcare students and professionals [19,22]. In general, self-compassionate attitudes (i.e., the opposite of self-critical attitudes) and mindfulness skills have been found to be strongly and negatively correlated with EA [32,33].

The main aim of the present study was to assess the psychometric properties of the AAQ-II in Brazilian PC providers using the Brazilian version of the questionnaire that was originally adapted [15]. Additionally, the study evaluates its convergence and divergence with respect to self-criticism and mindfulness and estimates its criterion validity on anxiety and depressive symptoms. We considered previous studies that have evaluated EA, self-criticism, and mindfulness as common potential mechanisms in MBPs [34] and past work that has studied the potential protective role of mindfulness, self-compassion, and psychological flexibility on burnout [19,22], which relates to anxiety/depressive symptomatology [35]. Thus, we expected: (1) strong positive relationships between EA and self-criticism; (2) strong negative relationships between EA and mindfulness skills; and (3) strong positive relationships between EA and anxiety and depressive symptoms. These explorations will help clarify the extent to which future research on EA may help enable actions and improvements in the well-being of PC workers and improve the quality of PC Brazilian services.

## 2. Materials and Methods

### 2.1. Design, Participants, and Procedure

A cross-sectional design was used with measurements obtained using online self-report techniques. A sample of PC providers from São Paulo, Brazil, who were employed between May and July 2015, were invited to participate. The sample size was estimated to exceed the recommended 10:1 ratio for the number of subjects to the number of test items [36], and it was thus established that around 100 participants would be needed at a minimum. Invitations were sent to 1600 potential participants via email, with follow-up emails sent three times in three weeks to achieve the greatest possible response ratio (RR).

A total of 407 participants responded to the study survey with no missing data and were therefore included in the analyses. The sociodemographic characteristics of the sample are summarized in Table 1. The average age of the sample was 41.09 years (SD = 10.09). The majority of the sample were female (84.5%), had a partner (70.3%), had children (60.7%), and were professionals who worked with a salary (61.2%). Physicians accounted for 17.7% of the sample, nurses accounted for 25.1%, and community health workers accounted for 57.2% of the sample. The average length of service was 17.19 years (SD = 9.81), with a mean of 5.47 years (SD = 5.53) at the current workplace. Most participants had permanent (90.4%) and full-time (89.9%) contracts. Nonetheless, 84.0% reported having economic difficulties to some extent. Participants worked for a mean of 39.25 h per week (SD = 26.81). Those who have had sick leave absences during the previous year (35.6%) showed an average of 19.0 sick leave days (SD = 51.83).

The participants gave their prior online informed consent, attesting to their willingness to participate. The ethics committee of the Federal University of São Paulo (UNIFESP; 26-10-2016; CAAE 30374114.1.0000.5505) approved this study.

### 2.2. Measurements

Sociodemographic information was collected regarding age; sex (male, female); relationships (with partner/married, with no partner); the number of children; professional category (volunteer, professional with a salary); job position (physician, nurse, community health worker); hours worked per week; length of service in years; years at the same job; contract period (temporary, permanent); contract type (full-time, part-time); economic difficulties (never to always); and sick leave in the past year (including days of sick leave).

The AAQ-II measures EA as the inability to act in the present moment, based on value-directed actions, when experiencing distressing psychological events. EA is, therefore, an example of psychological inflexibility [4]. The AAQ-II consists of seven items (e.g., “Emotions cause problems in my life”), and the items are responded to on a Likert-type scale (1 = ‘never’ to 7 = ‘always’). Higher scores indicate greater levels of EA. The Portuguese translation of the questionnaire, originally proposed to be used in Brazil [15], was used and evaluated in the present study.

The Self-Compassion Scale (SCS) [37] is a questionnaire that measures facets of self-compassion: ‘self-kindness’, ‘common humanity’, ‘mindfulness’, ‘self-judgement’, ‘isolation’, and ‘over-identification.’ The last three facets comprise a sub-group of negative factors called self-criticism [21] and include 13 items that assess how respondents perceive their actions towards themselves in difficult situations. These items are responded to on a Likert-type scale (1 = ‘almost never’ to 5 = ‘almost always’). Higher scores in the facets of self-judgment (e.g., “I try to see my failures as part of the human condition”), isolation (e.g., “When I’m feeling down, I tend to feel like most other people are happier than I am”), and over-identification (e.g., “when I’m feeling down, I tend to obsess and fixate on everything that is going wrong”) indicate greater self-criticism. It has been observed that the self-criticism items have better psychometric properties than the complete SCS questionnaire [21]. The Brazilian-validated version of the negative subscales of the SCS was used [21]. The internal consistency values in the present study were as follows: self-judgement (ω = 0.83), isolation (ω = 0.80), and over-identification (ω = 0.78).

The Five Facet Mindfulness Questionnaire (FFMQ) [38] is a 39-item questionnaire that measures mindfulness skills. Each item is scored on a Likert-type scale (1 = ‘never’ to 5 = ‘almost always’). These 39 items are divided into five mindfulness facets: ‘observing’ (i.e., noticing or attending to internal and external experiences such as sensations, thoughts, or emotions); ‘describing’ (i.e., labeling internal experiences with words); ‘acting with awareness’ (i.e., focusing on one’s activities at a given moment as opposed to behaving mechanically); ‘non-judging’ of inner experience (i.e., taking a non-evaluative stance towards thoughts and feelings); and ‘non-reactivity’ to inner experience (i.e., allowing thoughts and feelings to come and go without getting caught up in them or being carried away by them). The Brazilian-validated version of the FFMQ was used [39]. The FFMQ dimensions showed good internal consistency in the present study, with ω values between 0.70 (non-reactivity) and 0.87 (acting with awareness).

The Hospital Anxiety and Depression Scale (HADS) [40] is a 14-item questionnaire used to determine the levels of affective symptomatology and probable cases of anxiety and depression. It consists of two subscales (7 items each): HADS-Anxiety (e.g., “I feel tense or wound up”) and HADS-Depression (e.g., “I feel as if I am slowed down”). Each item can be scored from 0 to 3. The following cut-off points for each subscale have been proposed: ≤7 (no anxiety/depression); 8–10 (mild anxiety/depression); 11–14 (moderate anxiety/depression); and 15–21 (severe anxiety/depression). Both the HADS-Anxiety and HADS-Depression subscales have demonstrated sensitivity and specificity values of approximately 0.80 [41,42,43]. The Portuguese version of the HADS was used [41]. The internal consistency in the present study was anxiety (ω = 0.83) and depression (ω = 0.82).

### 2.3. Data Analyses

Sociodemographic data were described using means (SDs), or frequencies (percentages), according to the nature and distribution of each variable.

We described item characteristics by calculating means (SDs), skewness, kurtosis, item-rest correlations, and polychoric correlations. We checked whether the theoretically predicted unidimensional structure of the AAQ-II holds for Brazilian PC providers by using parallel analysis (PA) and confirmatory factor analysis (CFA). Parallel analysis (PA) was used to identify whether the number of factors to retain corresponded to the original one-factor solution. This was carried out by optimal implementation based on minimum rank factor analysis, generating 500 random correlation matrices [44]. With this procedure, a factor is significant if the associated eigenvalue is larger than that corresponding to a given percentile (95th of the distribution of eigenvalues derived from the random dataset). After testing the number of factors to retain, we developed a CFA and calculated standardized factorial weights and the percentage of explained variance in each item by means of communality values. The viability of the CFA model was tested using the comparative fit index (CFI), the Tucker–Lewis index (TLI), the root-mean-square error of approximation (RMSEA), and the weighted root-mean-square residual (SRMR). The CFI compares the fit of a target model to the fit of a null model (with values ≥0.95 for good and ≥0.90 for acceptable). The TLI is more sensitive to sample size (with values ≥0.95 for good and ≥0.90 for acceptable). The RMSEA is a measurement of the error of approximation to the population (with values ≤0.06 for good and ≤0.10 for acceptable). Lastly, the SRMR is the standardized difference between the observed and the predicted covariance (with values ≤0.06 for good and ≤0.10 for acceptable) [45,46].

The internal consistency of the scale was determined by calculating McDonald’s omega (ω) composite reliability, with coefficients ≥0.70 indicating adequate internal consistency values [47]. McDonald’s ω has the advantage of considering the strength of association between items and constructs and item-specific measurement errors, providing a more realistic estimate of true reliability than Cronbach’s alpha [48,49]. Mean inter-item polychoric correlations and mean item-rest correlations were also calculated.

The convergent/divergent validity of EA, self-criticism, and mindfulness skills was calculated using Pearson’s *r* correlation coefficient. Partial correlation coefficients were calculated to have a correlation estimate after controlling for the influence of the other factors (i.e., subscales of self-criticism and mindfulness). We also evaluated the extent to which self-criticism and mindfulness explained EA by constructing multiple linear regression models. The negative subscales of the SCS and FFMQ subscales were considered independent variables. At the same time, the latent value of the AAQ-II, calculated according to the Bayes expected a posteriori method (EAP) [50], was considered the dependent variable. The individual contribution of the independent variables in the multivariate model was estimated by standardized slope (beta) coefficients. The Wald test was used to evaluate its significance. The significance of the model’s explanatory power was examined by analysis of variance (ANOVA). The multiple correlation coefficient and the adjusted multiple determination coefficient were also estimated. Tolerance values were calculated to rule out possible collinearity problems; the Kolmogorov–Smirnov test was used to determine if residuals were normally distributed; and Durbin–Watson (DW) values were assessed to rule out autocorrelation problems in the error terms (criterion: DW ≈ 2.00) [51].

The criterion validity of EA on HADS-A and HADS-D was assessed by plotting receiving operating characteristics (ROC) curves comparing the AAQ-II with the HADS anxiety and depression subscales. The area under the curve (AUC) was calculated to represent the capacity of the AAQ-II to discriminate between cases and non-cases according to the states of anxiety and depression, applying the cut-off criteria aforementioned. Sensitivity and specificity values were used as case identification and non-case recognition measures. Positive and negative predictive values were also estimated to ascertain the capacity of the AAQ-II to detect true and false cases. Overall misclassification rates were calculated. The Youden Index was also calculated, which is unaffected by prevalence rates and represents the difference between the proportions of true cases and false cases identified, with a higher value indicating a more appropriate cut-off point.

The level of significance adopted in the tests was *p* < 0.05. Data analyses were carried out using the SPSS v28.0 (SPSS Inc., Chicago, IL, USA) and the Mplus v8.8 (Múthen & Múthen, Los Angeles, CA, USA) statistical software packages.

## 3. Results

### 3.1. Structural Validity of the Brazilian AAQ-II

Table 2 shows the descriptive and psychometric characteristics of the AAQ-II items. PA identified a one-factor solution, explaining 81.0% of the variance. The CFA model for the one-factor solution presented appropriate goodness of fit indices (CFI = 0.97; TLI = 0.95; RMSEA = 0.08 (90% CI = 0.05–0.010); SRMR = 0.03). The standardized weights of the items ranged from 0.74 to 0.88. The communality values of the items were ≥0.55. As the theoretical mean of each item considering the Likert-type scale used (ranging from 1 to 7) is four, empirical means were slightly below the theoretical mean.

### 3.2. Internal Consistency of the Brazilian AAQ-II

The McDonald’s omega composite reliability index showed a value of ω = 0.93. On the other hand, the mean of the item-rest values for the total sample was 0.77 (range: 0.72 to 0.83), and the mean inter-item polychoric correlation was 0.65. In conjunction, all this information suggested appropriate levels of internal consistency.

### 3.3. Convergence/Divergence and Explanatory Power of Self-Criticism and Mindfulness

Table 3 shows descriptive and Pearson correlations between the AAQ-II and self-criticism and mindfulness skills. Correlations indicated that the AAQ-II was strongly and positively associated with all the self-criticism sub-dimensions (e.g., self-judgement, isolation, and over-identification). In addition, the AAQ-II was also strongly and negatively related to acting with awareness and non-judging of the inner experience subscales of mindfulness. We also observed small-to-moderate negative correlations between the AAQ-II and the mindfulness facets of non-reactivity to inner experience and describing. The AAQ-II showed no significant relationships with the observing facet.

The explanatory power of the multiple regression model was very high (see Table 4), explaining 59% of the AAQ-II variance. The observing and describing facets of the FFMQ were the only subscales that did not significantly explain the AAQ-II variance in the multivariable model. The isolation facet of self-criticism was the factor that mostly explained AAQ-II variance, followed by the non-judging of the inner experience facet of the FFMQ, the over-identification self-criticism facet, the self-judgment self-criticism facet, the acting with awareness subscale of the FFMQ, and the non-reactivity facet of the FFMQ (demonstrating negative relationships with the FFMQ). The tolerance values ranged from 0.27 to 0.68, which suggests an acceptable degree of overlap among the self-criticism and mindfulness facets. The residuals of predicted values showed a normal distribution (*p* = 0.205), and the Durbin–Watson value was 2.05; thus, model assumptions were met, and the multiple linear regression analysis results could be interpreted with confidence.

### 3.4. Criterion Validity of the AAQ-II on Anxiety and Depressive Symptoms

Pearson correlations indicated that the AAQ-II was strongly and positively associated with anxiety (r = 0.56; *p* < 0.001) and depression (r = 0.55; *p* < 0.001) symptoms. As detailed in Table 5, the AAQ-II showed appropriate values of predictability when explaining different levels of anxiety and depressive symptoms. However, the sensitivity-specificity ratio was optimized when specifically considering severe states of anxiety or depression, meaning that the AAQ-II better predicted more severe symptoms.

## 4. Discussion

The main aim of this study was to assess the psychometric properties of the AAQ-II among Brazilian PC providers. Our findings suggest that the AAQ-II presents acceptable structural validity and internal consistency. The analyses yielded a one-factor solution, which is in line with the original theoretical proposal [4] and subsequent adaptations [52,53,54,55,56]. This solution explained more than 80% of the variance with appropriate goodness of fit indices. The internal consistency of the AAQ-II in terms of McDonalds’ omega was set at 0.93, which is very similar to the Cronbach’s alpha value reported in a previous validation study using Brazilian undergraduate students [9]. It is also aligned with other validation studies from different countries and populations [9,10,11,12,13,52,54,55,56,57]. Therefore, the Brazilian AAQ-II adaptation seems to be a structurally valid and internally consistent instrument to measure EA in Brazilian PC providers.

As a secondary objective, the present study aimed to assess the convergence/divergence of the AAQ-II with measures of self-criticism and mindfulness skills, as well as the capacity of self-criticism and mindfulness to explain EA using the AAQ-II. We have observed that both the absence of mindfulness skills and, largely, the presence of self-criticism was clearly associated with EA. Previous research has suggested that mindfulness skills, self-criticism (or its self-compassion counterpart), and EA may be playing a mediating role in third-wave cognitive psychological therapies, including those that incorporate meditation training [31]. Negative correlations between mindfulness skills and EA have been widely reported [58,59], which is supported by the theoretical framework of mindfulness. This suggests that mindfulness skills enhance attentional control, acceptance of one’s internal and external experiences, and non-reactivity to acute stressors [60]. However, as has been observed in the present study, the mindfulness facets of observing and describing may be less associated with psychological symptoms [61] and perhaps more dependent on previous meditation experience [62]. On the other hand, only a few studies have specifically identified self-criticism (along with its associated dimensions) as a construct that might be strongly related to EA. It is possible that being excessively self-critical during stressful life circumstances makes it difficult to implement emotion-regulation strategies, such as self-soothing, healthy reappraisal, and emotional acceptance [63]. Past work has supported a relationship between self-critical perfectionism and its predictive role in increasing EA [64]. The isolation factor from the self-criticism construct was more strongly associated with EA, which can be linked to previous findings that have found EA to be a significant mediator between emotion regulation abilities and loneliness [65]. In summary, EA, mindfulness skills, and self-criticism seem to be three key interrelated psychological processes called “third wave” CBTs, which place the focus primarily on the relationship between the individual and their internal experience (i.e., thoughts, emotions, and physical sensations) [66].

The third objective of the present study was to calculate the criterion validity of the AAQ-II on anxiety and depressive symptomatology. The results of the current paper propose potential cut-off points to have an indirect and, therefore, less stigmatizing measure of mental health status. We observed EA using the AAQ-II to be highly associated with anxiety and depressive symptoms, especially in the context of more severe symptoms. This result has been observed in different clinical contexts [58]. Longitudinal studies confirm that EA, as well as self-criticism, is a generalized vulnerability factor that confers transdiagnostic risk [67]. For example, exposure to trauma is an adverse life event closely linked with EA, which may increase the likelihood of aggravating symptoms or recurrence in the context of mental disorders [68]. EA is described as a “phenomenon that occurs when a person is unwilling to remain in contact with distressing experiences (e.g., bodily sensations, emotions, thoughts, memories, behavioral predispositions) and takes steps to alter the form or frequency of these events and the contexts that are associated with them” [69]. Individuals who tend to avoid uncomfortable or unwanted internal experiences are more vulnerable to stress because an avoidant response pattern makes it difficult to implement adaptive behaviors [69]. In line with our results, EA has been found to be a significant predictor of stress, anxiety, and depression [70,71]. As mentioned above, PC providers show an increased risk for job-related mood-disorder psychological distress [16], and evidence suggests that Brazilian PC workers are likely to develop anxiety or depression [17]. Further understanding of how PC workers respond to their internal experiences (i.e., thoughts, emotions, urges, sensations, memories, etc.) may provide important insights into the relevance of applying third-wave approaches (e.g., Mindfulness-Based Programs, Compassion-Based Programs, or Acceptance and Commitment Therapy) in this context. In general, our findings support the application of programs based on increasing mindfulness skills, reducing self-criticism, and enhancing psychological flexibility (by means of reducing EA) to improve anxiety and depression in Brazilian PC providers [72].

The findings of this study should be interpreted in light of the limitations. First, the initial study that used a Portuguese version of the AAQ-II in a sample of Brazilian college students [15] was scarce in terms of providing descriptive data at the item level. Thus, we were limited when comparing our results with those obtained in the previous study. Second, the included measures were all self-reported and, therefore, could be subject to desirability and recall bias. Thus, future research will need to use more objective measurements; for instance, clinical interviews. Third, due to this study’s cross-sectional nature, it is impossible to make causal inferences or establish temporal precedence. Fourth, although this study implemented email invitations to reach a wider audience, selection bias cannot be completely ruled out. In addition, we used a sample circumscribed to a specific Brazilian area, so the generalizability of results may be limited. Finally, the AAQ-II has been criticized for being saturated with personality traits or distress rather than specifically measuring EA [73,74], which could suppose an inflated association with anxiety and depressive symptoms, even with self-criticism [21]. Nevertheless, we ensured that there were no major violations of the basic assumptions required by the statistical models, increasing confidence in the results. Overall, the current paper addressed several knowledge gaps regarding the psychometric properties of the adapted Brazilian AAQ-II, the associations between mindfulness skills, self-criticism, and EA, and the criterion validity of this variable in relation to anxiety and depression in PC providers.

## 5. Conclusions

The AAQ-II is a suitable unidimensional test to measure EA in Brazilian PC providers. Our results provide further support for the association between mindfulness skills and EA and suggest that self-criticism, understood as the lack of self-compassion, also correlates significantly with EA. Moreover, EA was a clear predictor of anxiety and depression, particularly in the context of more severe symptoms. This highlights the importance of addressing EA when providing interventions to treat mental disorders.

## Figures and Tables

**Table 1 ijerph-20-00225-t001:** Sociodemographic characteristics of study participants.

Variables	
Age (years), *M* (*SD*)	41.09 (10.09)
Sex (female), *n* (%)	344 (84.5)
Relationship (partnership/married), *n* (%)	286 (70.3)
Number of children (none), *n* (%)	160 (39.3)
Category, *n* (%)	
Volunteer	158 (38.8)
Professional with a salary	249 (61.2)
Job position, *n* (%)	
Physician	72 (17.7)
Nurse	102 (25.1)
CHW	233 (57.2)
Hours worked per week, *M* (*SD*)	39.25 (26.81)
Length of service (years), *M* (*SD*)	17.19 (9.81)
Years at the same job, *M* (*SD*)	5.47 (5.53)
Contract period, *n* (%)	
Temporary	39 (9.6)
Permanent	368 (90.4)
Contract type, *n* (%)	
Full-time	366 (89.9)
Part-time	41 (10.1)
Economic difficulties, *n* (%)	
Never	65 (16.0)
Sometimes	153 (37.6)
Almost always	112 (27.5)
Always	77 (18.9)
Sick leave in the past year (yes), *n* (%)	145 (35.6)

Note: CHW: community health workers.

**Table 2 ijerph-20-00225-t002:** Descriptive and psychometric characteristics of the AAQ-II items (*n* = 407).

Items	Mean (SD)	Skewness	Kurtosis	Item-Rest	λ	h^2^
Item 1	3.21 (1.75)	0.53	−0.62	0.79	0.83	0.70
Item 2	3.11 (1.75)	0.56	−0.58	0.82	0.85	0.73
Item 3	3.59 (1.81)	0.27	−0.87	0.77	0.81	0.66
Item 4	3.01 (1.84)	0.65	−0.63	0.83	0.88	0.77
Item 5	3.55 (1.70)	0.27	−0.63	0.73	0.75	0.56
Item 6	3.40 (1.79)	0.33	−0.87	0.72	0.74	0.55
Item 7	3.57 (1.72)	0.21	−0.83	0.75	0.77	0.59

Note: λ = CFA standardized regression weights. h^2^ = communality values.

**Table 3 ijerph-20-00225-t003:** Descriptive and correlations between the AAQ-II and self-criticism/mindfulness.

Scales	Mean (SD)	AAQ-II (r)
AAQ-II (range: 7 to 49; theoretical average: 28)	23.43 (10.34)	
Self-judgement (range: 5 to 25; theoretical average: 15)	16.10 (4.14)	0.62 *
Isolation (range: 4 to 20; theoretical average: 12)	11.81 (3.77)	0.66 *
Over-identification (range: 4 to 20; theoretical average: 12)	12.51 (3.50)	0.69 *
Observing (range: 5 to 40; theoretical average: 24)	24.28 (7.41)	−0.07
Describing (range: 5 to 40; theoretical average: 24)	26.98 (6.60)	−0.27 *
Acting aware (range: 5 to 40; theoretical average: 24)	21.73 (6.89)	−0.47 *
Non-judging (range: 5 to 40; theoretical average: 24)	17.57 (7.10)	−0.52 *
Non-reactivity (range: 5 to 35; theoretical average: 21)	18.99 (4.98)	−0.18 *

Note: r = Pearson’s correlations. * *p* < 0.001.

**Table 4 ijerph-20-00225-t004:** Explanatory power of self-compassion and mindfulness facets on AAQ-II.

Multivariable Model	R_y.123_	Adj-R^2^_y.123_	DW	*p* ^a^
**Independent variables**	0.77	0.59 ***	2.05	0.370
	**T**	**R_y3.12_**	**Beta**	** *p* ^b^ **
Self-judgement	0.40	0.15	0.15	0.004
Isolation	0.44	0.31	0.31	<0.001
Over-identification	0.27	0.14	0.17	0.007
Observing	0.55	<0.01	<0.01	0.983
Describing	0.60	−0.06	−0.05	0.220
Acting aware	0.68	−0.17	−0.14	<0.001
Non-judging	0.61	−0.22	−0.19	<0.001
Non-reactivity	0.62	−0.13	−0.11	0.008

Note: R_y.123_ = multiple correlation coefficient. Adj-R^2^_y.123_ = adjusted multiple determination coefficient. *** = *p* < 0.001 (*p*-value associated with regression). DW = Durbin–Watson. *p*
^a^ = Kolmogorov-Smirnoff normality contrast on residuals. T = tolerance. R_y3.12_ = partial correlation coefficient. Beta = standardized slope. *p*
^b^ = *p*-value of Wald test on standardized slopes.

**Table 5 ijerph-20-00225-t005:** Criterion validity coefficients of AAQ-II on HADS-anxiety and HADS-depression.

**Anxiety**	**Mild/Moderate/Severe** **(AAQ-II > 20)**		**Moderate/Severe** **(AAQ-II > 24)**		**Severe** **(AAQ-II > 27)**	
	**Index**	**95%CI**	**Index**	**95%CI**	**Index**	**95%CI**
AUC	0.78	0.73–0.82	0.76	0.71–0.81	0.85	0.80–0.90
SEN	0.72	0.67–0.78	0.70	0.62–0.77	0.78	0.64–0.92
SPE	0.69	0.61–0.76	0.71	0.66–0.75	0.72	0.68–0.77
PPV	0.78	0.72–0.83	0.60	0.52–0.67	0.24	0.16–0.32
NPV	0.62	0.54–0.69	0.79	0.74–0.85	0.97	0.94–0.99
OMR	0.29	0.25–0.44	0.30	0.25–0.35	0.27	0.23–0.31
YIN	0.41	0.32–0.51	0.40	0.31–0.50	0.50	0.37–0.64
**Depression**	**mild/moderate/severe** **(AAQ-II > 23)**		**moderate/severe** **(AAQ-II > 26)**		**severe** **(AAQ-II > 30)**	
	**Index**	**95%CI**	**Index**	**95%CI**	**Index**	**95%CI**
AUC	0.78	0.73–0.82	0.76	0.70–0.82	0.87	0.80–0.94
SEN	0.74	0.67–0.81	0.70	0.60–0.81	0.84	0.68–1.00
SPE	0.71	0.65–0.77	0.70	0.65–0.75	0.82	0.78–0.86
PPV	0.65	0.58–0.72	0.40	0.32–0.48	0.23	0.14–0.32
NPV	0.79	0.74–0.85	0.90	0.86–0.94	0.99	0.97–1.00
OMR	0.28	0.23–0.32	0.30	0.25–0.34	0.18	0.14–0.22
YIN	0.45	0.36–0.54	0.41	0.30–0.59	0.66	0.51–0.81

Note: “mild/moderate/severe” = no (anxiety or depression) vs. mild/moderate/severe. “moderate/severe” = no (anxiety or depression)/mild vs. moderate/severe. “severe” = no (anxiety or depression)/mild/moderate vs. severe. Cut-offs for the HADS are specified in the measurements section. AUC = area under the curve. SEN = sensitivity. SPE = specificity. PPV = positive predictive value. NPV = negative predictive value. OMR = overall misclassification rate. YIN = Youden index.

## Data Availability

The original datasets analyzed during the current study are available from the corresponding author upon reasonable request.

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
