# Peer review of "Experiential Avoidance in Primary Care Providers: Psychometric Properties of the Brazilian “Acceptance and Action Questionnaire” (AAQ-II) and Its Criterion Validity on Mood Disorder-Related Psychological Distress"

_ijerph, 2022, doi:10.3390/ijerph20010225_

Round 1

Reviewer 1 Report

The article gives the impression of a research report rather than a scientific article. Both the presentation of the background in the introduction and the discussion are very limited and, in my opinion, need to be significantly expanded.

With this number of Authors, it is somewhat surprising that there is no adequate introduction to the topic of PF and its correlates, no deeper thoughtful discussion section and no reference to a wider number of relevant references demonstrating the Authors' broader knowledge and understanding of such an important topic.

With the corrections made, the paper has a chance to become a very interesting article.

Author Response

Thank you very much for your comments and observations. We completely agree in that the manuscript needed to be expanded on and more details needed to be  provided in the background and discussion sections. As can be seen now in the new version of the manuscript, we have completed the introduction with more specificity and have provided more information, so that the main psychological constructs of interest and the relationship between them are more clearly outlined and adequately referenced. In addition, the objective of the research has been contextualized in more detail, with additional information on the background and current situation of the AAQ-II questionnaire to justify the work carried out in this paper. We have also introduced more thoughtful comments in the discussion section, with the corresponding relevant references to help explain and support  interpretations. In general, we have expanded the whole manuscript to be an  adequate and balanced length to be  a scientific article. 

Reviewer 2 Report

The article draft “Psychological Flexibility in Primary Care Providers: Psychometric properties of the Brazilian ‘Acceptance and Action Questionnaire’ (AAQ-II) and its explanatory ability on mood disorder-related psychological distress” evaluates the psychometric properties of the AAQ-II among PC Brazilian providers.

Overall, I think the article tries to answer a straightforward research question, but seems unnecessarily difficult to understand. For me, it was not clear which analyses were undertaken for what purposes. My advice would be to reduce the confusing multiplicity of analyses to what is inherently necessary to answer the research questions. I will go into further details in the following.

In the introduction section, I think the objectives and motivation for the current study should be explained more clearly. If the authors state that the AAQ-II “has shown questionable internal consistency in populations with specific features, and content-specific versions have been created to increase psychometric properties in target populations”, readers may ask which features are meant and, more important, what is the target population? Is the target popuation specific (i.e., employees in medical service)? The cited publication [6] is spanish, so it might be worthwhile to give a short summary. An illustrating example, which content-specific adaptation have been made to overcome which [methodological] problems, would be helpful.

In section 2.2, authors introduce SCS, FFMQ, and HADS. However, the reader does not know why these scales are necessary for evaluating AAQ-II. Later in section 2.3, it is mentioned that the SCS scales self-criticism and mindfulness are used to evaluate convergent/divergent validity of EA. However, prior to section 2.2, authors should explain to the reader which correlations/interrelations with SCS, FFMQ etc. they expect and why, for example according to which kind of validity. Moreover, if other studies have examined correlations of AAQ-II with some SCS scales for specific populations, the results should be reviewed briefly.

In section 2.3, I do not understand why an EFA is undertaken. If the AAQ-II consists of 7 items measuring EA, a simple uni-dimensional CFA might be sufficient to check whether the theoretically predicted structure holds for Brazilian PC providers. Especially if the AAQ-II was already evaluated in other subpopulations, I think it might be more relevant whether the uni-dimensional structure which had been verified for other populations also holds for Brazilian PC providers.

At least for myself, I can say that I’m not familiar with Mardia’s coefficient or KMO index. Authors might provide short explanations or literature references for these methods.

Section 2.3 is very confusing to me. For example, the authors state: “The convergent/divergent validity of EA, self-criticism and mindfulness skills was calculated by applying Pearson’s r correlation coefficient. We also evaluated the capacity of self-criticism and mindfulness to explain EA by constructing multiple linear regression models.” Already in the introduction/theory section, authors should explain which correlation coefficient they expect if convergent/divergent validity of EA holds (something like “expect moderate positive/negative correlation”). Otherwise, readers do not understand which consequences follows from the results according to the research questions.

It is hard to understand what follows from section 3.1 or table 1. Authors enumerate many indices (Mardia’s, KMO, Bartlett, …) but miss to provide valuable interpretation. It is well understandable if PA supports one factor solution, but what do we learn from the further indices? Authors should provide more comprehensible explanation. In my (naïve) view, many of the indices seem dispensable without losing the main conclusion.

Minor issues:

  • Language: the authors write “PC Brazilian providers”. I think, “Brazilian PC providers” is more appropriate
  • In table 4, the relation of AAQ-II boundaries (mild, moderate, severe) is not clear. In lines 126-129, the boundaries/cu-off points for the HADS are described, but not for the AAQ-II.  
  • Some confusion occurs, for example in 222-223: “Only ‘observing’ and ‘describing’ did not contribute significantly to explain the AAQ-II; ‘isolation’ was the factor that explained EA more.” Are AAQ-II and EA used interchangeably here?
  • GFI is missing in table 2 (it only occurs in the notes)
  • In table 3, I wonder why the adjusted R2 is negative for Non-reacting.

Author Response

Reviewer 2

The article draft “Psychological Flexibility in Primary Care Providers: Psychometric properties of the Brazilian ‘Acceptance and Action Questionnaire’ (AAQ-II) and its explanatory ability on mood disorder-related psychological distress” evaluates the psychometric properties of the AAQ-II among PC Brazilian providers.

Overall, I think the article tries to answer a straightforward research question, but seems unnecessarily difficult to understand. For me, it was not clear which analyses were undertaken for what purposes. My advice would be to reduce the confusing multiplicity of analyses to what is inherently necessary to answer the research questions. I will go into further details in the following.

Answer:

Thank you for these constructive comments. As can be seen in the new version of the manuscript, we have specified the research aims with more clarity, which has allowed us to clarify which data analyses were undertaken. We have simplified the number of analyses to what we consider is necessary to be able to answer our research questions.

In the introduction section, I think the objectives and motivation for the current study should be explained more clearly. If the authors state that the AAQ-II “has shown questionable internal consistency in populations with specific features, and content-specific versions have been created to increase psychometric properties in target populations”, readers may ask which features are meant and, more important, what is the target population? Is the target popuation specific (i.e., employees in medical service)? The cited publication [6] is spanish, so it might be worthwhile to give a short summary. An illustrating example, which content-specific adaptation have been made to overcome which [methodological] problems, would be helpful.

Answer:

Many thanks for this constructive comment. As can be seen in the new version of the manuscript, we have now revised the introduction and have provided  more information so that the objective for the current study is clearly  explained and contextualized. We have justified why Brazilian Primary Care providers constitute the study target population (page 2, lines 77-85; “Primary Care (PC) providers constitute a specific target population, since they are highly affected by job-related mood-disorder psychological distress [16]. In Brazil, it has been highlighted that 21% of PC workers might suffer from common mental disorders, such as anxiety and depression [17]. PC providers could be using EA as an escape mechanism to cope with their sense of lack of control over stressors in their current working environments [18]. In fact, it has been proposed that, for example, EA is a significant predictor of burnout among psychology and nursing undergraduate students [19]”). We have also added information on the background and current situation of the AAQ-II questionnaire to justify the work undertaken in the current paper. Moreover, examples of different AAQ-II versions (used for different contexts and populations) are provided to outline the psychometric problems (e.g., questionable internal consistency) that justify the objective of the current work (page 2, lines 54-78). Given the weaknesses of the original Brazilian AAQ-II validation study, the present study provides more rigorous and appropriate analyses to evaluate the psychometric properties of the scale. To address these shortcomings, we have included confirmatory factor analysis and its corresponding fit indices and have provided internal consistency estimates according to the most recent standards. The limitations of the original validation study include limited external validity, only using exploratory factorial analysis (EFA), not making use of confirmatory factorial analysis (CFA), not evaluating  the fit of the model proposed, and not providing internal consistency estimates according to recent standards (page 2, lines 71-78).

In section 2.2, authors introduce SCS, FFMQ, and HADS. However, the reader does not know why these scales are necessary for evaluating AAQ-II. Later in section 2.3, it is mentioned that the SCS scales self-criticism and mindfulness are used to evaluate convergent/divergent validity of EA. However, prior to section 2.2, authors should explain to the reader which correlations/interrelations with SCS, FFMQ etc. they expect and why, for example according to which kind of validity. Moreover, if other studies have examined correlations of AAQ-II with some SCS scales for specific populations, the results should be reviewed briefly.

Answer:

Thanks for this useful comment. We have now included in the introduction section the concepts of self-criticism and mindfulness (page 2-3, lines 86-107) as important psychological constructs of interest that are associated with physical and mental health. The following section has been added to provide this detail: “Two other variables have been identified as significantly associated with distress and burnout in similar populations: self-criticism, and mindfulness [20–22]. Self-criticism, as the negative counterpart of self-compassion [21], can be considered as a dysfunctional emotion regulation strategy [23,24] used to deal with negative events in which one views their own mistakes as the cause of an event [25]. On the other hand, mindfulness refers to a mental trait or state that is associated with increased awareness with the present moment and characterized by a non-judgmental attitude. Past research has indicated that mindfulness skills are associated with several indicators of physical and psychological health [26,27]”. In addition, the relationships between self-criticism, mindfulness, and EA (as an example of psychological inflexibility) are now more clearly presented. In this sense, we have explained in the introduction that “Reducing self-criticism and improving mindfulness skills may help prevent and reduce psychological distress through the process of breaking patterns of reactivity that trigger anxiety and depression and the cycle of suffering [28]. Both the presence of self-criticism and the absence of mindfulness skills can be addressed through the practice of meditative exercises that promote attention to the present moment [29]. It has been observed that mindfulness practices, for example in the context of mindfulness-based programs (MBPs), may improve anxiety, depression, psychological distress, and mental well-being in healthcare professionals [30]. Moreover, past work suggests that these MBPs may obtain improvements in outcome through the mediating role of EA, self-criticism, and mindfulness [31]. Previous studies have shown significant relationships between burnout, EA, and self-criticism, as well as significant and negative relationships between burnout and the presence of mindfulness skills in healthcare students and professionals [19,22]. In general, self-compassionate attitudes (i.e., the opposite of self-critical attitudes) and mindfulness skills have been found to be strongly and negatively correlated with EA [32,33]”. We have also expanded the description of the instruments used so that readers have more information on how to interpret the constructs (pages 4-5, lines 149-198).

In section 2.3, I do not understand why an EFA is undertaken. If the AAQ-II consists of 7 items measuring EA, a simple uni-dimensional CFA might be sufficient to check whether the theoretically predicted structure holds for Brazilian PC providers. Especially if the AAQ-II was already evaluated in other subpopulations, I think it might be more relevant whether the uni-dimensional structure which had been verified for other populations also holds for Brazilian PC providers.

Answer:

Thanks for this helpful comment. We are in agreement with your proposal and we have removed the EFA and have developed a CFA using the whole sample. In addition, we have now used more recent standards in order to evaluate the fit of the model (e.g., CFI, TLI, RMSR, RMSEA). Please, see the data analyses and results sections for more detail.

At least for myself, I can say that I’m not familiar with Mardia’s coefficient or KMO index. Authors might provide short explanations or literature references for these methods.

Answer:

The mentioned indices are mainly used to provide an idea of the multivariable distribution of the items, as well as the characteristics of the correlation matrix when applying EFA. As requested by the reviewer, we ended up removing this analysis and have used polychoric correlations for the CFA in order to overcome any potential problems derived from the use of ordinal variables (items). Therefore, as a result, the aforementioned indices are no longer necessary and have been removed.

Section 2.3 is very confusing to me. For example, the authors state: “The convergent/divergent validity of EA, self-criticism and mindfulness skills was calculated by applying Pearson’s r correlation coefficient. We also evaluated the capacity of self-criticism and mindfulness to explain EA by constructing multiple linear regression models.” Already in the introduction/theory section, authors should explain which correlation coefficient they expect if convergent/divergent validity of EA holds (something like “expect moderate positive/negative correlation”). Otherwise, readers do not understand which consequences follows from the results according to the research questions.

Answer:

Thank you for this constructive comment. We have clarified what we expect in terms of convergent and divergent validity in the context of our study aim and we have included the following section to clarify this:  “The main aim of the present study was to assess the psychometric properties of the AAQ-II in PC Brazilian providers, using the Brazilian version of the questionnaire that was originally adapted [15]. Additionally, the study evaluates its convergence and divergence with respect to self-criticism and mindfulness and estimates its criterion validity on anxiety and depressive symptoms. In light of previous studies that have evaluated EA, self-criticism, and mindfulness as common potential mechanisms in MBPs [34], and past work that has studied the potential protective role of mindfulness, self-compassion and psychological flexibility on burnout [19,22], which relates to anxiety/depressive symptomatology [35], we expected: 1) strong positive relationships between EA and self-criticism; 2) strong negative relationships between EA and mindfulness skills; and c) strong positive relationships between EA and anxiety and depressive symptoms. These explorations will help clarify the extent to which future research on EA may help enable actions and improvements in the well-being of PC workers, and in the context of improving the quality of PC Brazilian services” (page 3, lines 108-121).

It is hard to understand what follows from section 3.1 or table 1. Authors enumerate many indices (Mardia’s, KMO, Bartlett, …) but miss to provide valuable interpretation. It is well understandable if PA supports one factor solution, but what do we learn from the further indices? Authors should provide more comprehensible explanation. In my (naïve) view, many of the indices seem dispensable without losing the main conclusion.

 Answer:

Thanks for this constructive comment. We have now specified the research aims, which has allowed us to better explain in the data analysis section which analyses were undertaken to address these questions. We have simplified the number of analyses to what we consider is necessary to be able to answer our research questions. In this sense, we have removed Mardia’s, KMO, Bartlett, and the entire EFA, so that Table 1 is now easier to interpret. In addition, we have now used more recent standards in order to evaluate the fit of the model (e.g., CFI, TLI, RMSR, RMSEA), and the meaning of the corresponding indices have now been explained in the data analyses section (pages 5-6, lines 199-257). For example, we have included: “The CFI compares the fit of a target model to the fit of a null model (with values ≥ 0.95 for good and ≥ 0.90 for acceptable). The TLI is more sensitive to sample size (with values ≥ 0.95 for good and ≥ 0.90 for acceptable). The RMSEA is a measurement of the error of approximation to the population (with values ≤ 0.06 for good and ≤ 0.10 for acceptable). Lastly, the SRMR is the standardized difference between the observed and the predicted covariance (with values ≤ 0.06 for good and ≤0 .10 for acceptable) [45,46]”. In addition, we have removed the previous Table 2 and the different models of reliability (e.g., congeneric, Tau-equivalent, Parallel), as well as their corresponding fit indices, and have provided the reliability value by using McDonalds’s omega. We have also included a new table with the simple bivariate correlations between the AAQ-II and the self-criticism facets and mindfulness skills (Table 3) to facilitate understanding. We have also simplified the presentation of the multivariable analysis (multiple regression) described in Table 4 by removing the F-test values, unstandardized slopes, and standard errors from the table. We also removed some of the indices that were previously provided in Table 5 (e.g., PLR=positive likelihood ratio, NLR=negative likelihood ratio) so that we are more focused on the indispensable ones, all of which are now explained in detail in the data analyses section to facilitate interpretation (pages 5-6, lines 199-257). Following from this revised analysis section we have also clarified the results section, which we now believe is more understandable (pages 4-6). Thank you for this valuable suggestion.

Minor issues:

Language: the authors write “PC Brazilian providers”. I think, “Brazilian PC providers” is more appropriate

Answer:

Thanks for this suggestion. We have corrected it throughout the whole manuscript.

In table 4, the relation of AAQ-II boundaries (mild, moderate, severe) is not clear. In lines 126-129, the boundaries/cutoff points for the HADS are described, but not for the AAQ-II. 

Answer:

Thanks for this comment. We have now clarified the boundaries of anxiety and depression used to evaluate the criterion validity of the AAQ-II in Table 5. The boundaries for anxiety and depression are as follows: “mild/moderate/severe” = no (anxiety or depression) vs. mild/moderate/severe. “moderate/severe” = no (anxiety or depression)/mild vs. moderate/severe. “severe” = no (anxiety or de-pression)/mild/moderate vs. severe. Cut-offs for the HADS are specified in the measurements section.

The cutoff points for the AAQ-II have been established, as a result of the criterion validity analyses, and are provided in the Table 5. For anxiety: AAQ-II>20, AAQ-II>24, and AAQ-II>27. For depression: AAQ-II>23, AAQ-II>26, and AAQ-II>30. To highlight this, we have also included the following in the results section:  “The results of the current paper propose potential cut-off points to have an indirect, and therefore less stigmatizing, measure of mental health status” (page 9, lines 356-358).

Some confusion occurs, for example in 222-223: “Only ‘observing’ and ‘describing’ did not contribute significantly to explain the AAQ-II; ‘isolation’ was the factor that explained EA more.” Are AAQ-II and EA used interchangeably here?

Answer:

Thanks for this comment. We have now corrected this, and AAQ-II is being consistently used throughout the results section.

GFI is missing in table 2 (it only occurs in the notes)

Answer:

Thanks for this comment. Following the reviewer’s suggestion above, we have now removed Table 2 in order to simplify the analyses and indices provided to include only those that are indispensable (e.g., now reliability is provided using McDonalds’s omega).

In table 3, I wonder why the adjusted R2 is negative for Non-reacting.

Answer:

Thanks for this. It has now been corrected. As can be seen now in Table 4, the first line of the table provides the indices of the multivariable model, while the rest of the lines provide the indices for the independent variables. We have now highlighted this by separating these two parts of the table (Multivariable model, Independent variables).

Round 2

Reviewer 2 Report

Thanks for the revised version, I think the manuscript has improved substantially. I see only some minor issues which might be solved by the authors prior to publication:

In table 2, authors provide descriptive characteristics of the 7 AAQ-II items. Authors might complement that, with a likert scale ranging from 1 to 7, the theoretical mean is 4, meaning that empirical means are slightly below the theoretical mean. It would be interesting whether these values are smaller or larger than in different comparison groups, for example the Brazilian college students from reference [15].

Likewise, in table 3, the means are not very informative, because numerical values depend on the scale (5- or 7-point likert) as well as on the number of items. For the AAQ-II, for example, it is not clear whether 23.43 is much or little. Authors might add [perhaps in the table] that with 7 likert-type items, minimal sum score is 7, maximum sum score is 49, and the theoretical average is 4*7=28. Readers than may see that the empirical mean is slightly below the theoretical mean -- the difference 28-23.43, however, is less than one standard deviation.

Authors might provide references for Durbin-Watson and MacDonalds Omega. For the second one, Zinbarg et al. (2005) might be appropriate:

Zinbarg, R. E., Revelle, W., Yofel, I., & Li, W. (2005). Cronbach's α, Revelle's β, and McDonald's ωH: Their relations with each other and two alternative conceptualizations of reliability. Psychometrika, 70(1), 123-133. https://doi.org/10.1007/s11336-003-0974-7

Line 214: "the Tucker–Lewis index (TLI), The root ..." The second "the" should be written in lower case.

Author Response

Thanks for the revised version, I think the manuscript has improved substantially. I see only some minor issues which might be solved by the authors prior to publication:

Answer: Many thanks for this comment. This has been possible thanks to the reviewer’s suggestions, so we are very grateful for that.

In table 2, authors provide descriptive characteristics of the 7 AAQ-II items. Authors might complement that, with a likert scale ranging from 1 to 7, the theoretical mean is 4, meaning that empirical means are slightly below the theoretical mean. It would be interesting whether these values are smaller or larger than in different comparison groups, for example the Brazilian college students from reference [15].

Answer: Thanks for this relevant suggestion. We have now included in the results section (page 6, line 264) that: “As the theoretical mean of each item considering the Likert-type scale ranging from 1 to 7 is four, empirical means were slightly below the theoretical mean”. Descriptive data at the item level was not provided in the initial Brazilian validation study [15], which was very limited in this sense, and thus we were not able to compare our results with the ones obtained in the mentioned study. We have therefore included in the limitations section of the manuscript that (page 9, lines 384-387): “the initial study that used a Portuguese version of the AAQ-II in a sample of Brazilian college students [15] was scarce in terms of providing descriptive data at the item level, and thus, we were limited when comparing our results with those obtained in the previous study”.

15. Barbosa, L.M.; Murta, S.G. Propriedades Psicométricas Iniciais Do Acceptance and Action Questionnaire - II - Versão Brasileira. Psico-USF 2015, 20, 75–85, doi:10.1590/1413-82712015200107.

Likewise, in table 3, the means are not very informative, because numerical values depend on the scale (5- or 7-point likert) as well as on the number of items. For the AAQ-II, for example, it is not clear whether 23.43 is much or little. Authors might add [perhaps in the table] that with 7 likert-type items, minimal sum score is 7, maximum sum score is 49, and the theoretical average is 4*7=28. Readers than may see that the empirical mean is slightly below the theoretical mean -- the difference 28-23.43, however, is less than one standard deviation.

Answer: Thanks for this comment. We have now included in the Table 3 the range and theoretical average of each scale, so that readers are in better conditions in order to interpret properly the descriptive data that is provided in the referred table.

Authors might provide references for Durbin-Watson and MacDonalds Omega. For the second one, Zinbarg et al. (2005) might be appropriate:

Zinbarg, R. E., Revelle, W., Yofel, I., & Li, W. (2005). Cronbach's α, Revelle's β, and McDonald's ωH: Their relations with each other and two alternative conceptualizations of reliability. Psychometrika, 70(1), 123-133. https://doi.org/10.1007/s11336-003-0974-7

Answer: Thanks for this comment, we have now included the following references for Durbin-Watson and MacDonalds Omega, as suggested by the reviewer. We hope that these new inclusions have enhanced your view of the appropriateness of the references used:

Zinbarg, R. E., Revelle, W., Yofel, I., & Li, W. (2005). Cronbach's α, Revelle's β, and McDonald's ωH: Their relations with each other and two alternative conceptualizations of reliability. Psychometrika, 70(1), 123-133. https://doi.org/10.1007/s11336-003-0974-7

Dodge, Y. Durbin–Watson Test. In: The Concise Encyclopedia of Statistics. The Concise Encyclopedia of Statistics. Springer, New York, NY; 2008. https://doi.org/10.1007/978-0-387-32833-1_122

Line 214: "the Tucker–Lewis index (TLI), The root ..." The second "the" should be written in lower case.

Answer: Thanks for spotting this. It has now been corrected.
